# Minimally Invasive Free-Breathing Gating-Free Extracellular Cellular Volume Quantification for Repetitive Myocardial Fibrosis Evaluation in Rodents

**DOI:** 10.3390/biom15121732

**Published:** 2025-12-12

**Authors:** Devin Raine Everaldo Cortes, Thomas Becker-Szurszewski, Sean Hartwick, Muhammad Wahab Amjad, Soheb Anwar Mohammed, Xucai Chen, John J. Pacella, Anthony G. Christodoulou, Yijen L. Wu

**Affiliations:** 1Department of Pediatrics, School of Medicine, University of Pittsburgh, Pittsburgh, PA 15224, USA; drc76@pitt.edu (D.R.E.C.); thb72@pitt.edu (T.B.-S.); hartwick@pitt.edu (S.H.); 2Department of Bioengineering, Swanson School of Engineering, University of Pittsburgh, Pittsburgh, PA 15261, USA; 3Animal Imaging Core, Rangos Research Center, UPMC Children’s Hospital of Pittsburgh, Pittsburgh, PA 15224, USA; 4Center for Ultrasound Molecular Imaging and Therapeutics, Heart and Vascular Medicine Institute, University of Pittsburgh, Pittsburgh, PA 15213, USA; mua56@pitt.edu (M.W.A.); som57@pitt.edu (S.A.M.); chenxx2@upmc.edu (X.C.); pacellajj@upmc.edu (J.J.P.); 5Department of Radiological Sciences, David Geffen School of Medicine, University of California Los Angeles, Los Angeles, CA 90095, USA; achristodoulou@mednet.ucla.edu; 6Department of Bioengineering, Samueli School of Engineering, University of California Los Angeles, Los Angeles, CA 90095, USA

**Keywords:** extracellular volume (ECV), fibrosis, rodent

## Abstract

**Background**: Interstitial myocardial fibrosis is a crucial pathological feature of many cardiovascular disorders. Myocardial fibrosis resulting in extracellular volume (ECV) expansion can be quantified via cardiac MRI (CMR) with T_1_ mapping before and after minimally invasive gadolinium (Gd) contrast agent administration. However, longitudinal repetitive ECV measurements are challenging in rodents due to the prolonged scan time with cardiac and respiratory gating that is required for conventional T_1_ mapping and the invasive nature of the rodent intravenous lines. **Methods**: To address these challenges, the objective of this study is to establish a fast, free-breathing, and gating-free ECV procedure using a minimally invasive subcutaneous catheter for in-scanner Gd administration that can allow longitudinal repetitive ECV evaluations in rodent models. This is achieved by the (1) IntraGate sequence for free-breathing, gating-free cardiac imaging; (2) minimally invasive subcutaneous in-scanner Gd administration; and (3) fast T_1_ mapping with a varied flip angle (VFA) in conjunction with (4) triple jugular vein blood T_1_ normalization. Additionally, full cine CMR (multi-slice short-axis, long-axis 2-chamber, and long-axis 4-chamber) was acquired during the waiting period to assess comprehensive cardiac function and strain. **Results**: We successfully established a minimally invasive fast ECV quantification protocol to enable longitudinal repetitive ECV quantifications in rodents. Minimally invasive subcutaneous Gd bolus administration induced a reasonable dynamic contrast enhancement (DCE) time course, reaching a steady state in ~20 min for stable T_1_ quantification. The free-breathing gating-free VFA T_1_ quantification scheme allows for rapid cardiac (~2.5 min) and jugular vein (49 s) T_1_ quantification with no motion artifacts. The triple jugular vein T_1_ acquisitions (1 pre-contrast and 2 post-contrast) immediately flanking the heart T_1_ acquisitions enable accurate myocardial ECV quantification. Our data demonstrated that left-ventricular myocardial ECV quantification was highly reproducible with repeated scans, and the ECV values (0.25) are comparable to reported ranges among humans and rodents. This protocol was successfully applied to the ischemia–reperfusion injury model to detect myocardial fibrosis, which was validated by histopathology. **Conclusions**: We established a simple, fast, minimally invasive, and robust CMR protocol in rodents that can enable longitudinal repetitive ECV quantification for cardiovascular disease progression. It can be used to monitor disease regression with interventions.

## 1. Introduction

Interstitial myocardial fibrosis [1,2,3] is a crucial hallmark of cardiovascular diseases, such as hypertrophic cardiomyopathy (HCM) [4,5,6], congenital heart disease (CHD) [7,8,9], and heart failure with preserved ejection fraction (HFpEF) [10,11,12,13,14]. Myocardial fibrosis involves complex interactions across several biomolecules, such as extracellular matrix (ECM) proteins, cytokines, and regulatory mRNAs, that can contribute to fibrotic remodeling [2,3]. Myocardial fibrosis resulting in extracellular volume (ECV) expansion can be quantified by cardiac MRI (CMR) with T_1_ mapping before and after minimally invasive gadolinium (Gd) contrast agent administration [15,16,17,18,19,20,21,22,23]. As ECV expansion is progressive [14] and reversible, it is a therapeutic target and can be a surrogate biomarker for therapeutic trials [1,24].

Animal models are indispensable for modeling human diseases to gain a mechanistic understanding and testing therapeutic efficacy [25,26,27]. Rodent models are suitable for cardiovascular research [25,28,29] because rodents and humans share similar anatomy with 4-chamber hearts and similar great arteries and veins, comparable physiology, and conserved biological processes, genes, gene modifiers, and molecular pathways [25,30,31,32]. Compared to large animals, rodents have much shorter breeding cycles (18–21 days) and weaning ages (21 days), facilitating time-efficient studies and cost reduction for animal housing. More importantly, the availability of inbred lines and transgenic strains makes mechanistic investigations of genetic and epigenetic regulations possible.

However, longitudinal repetitive ECV measurements are challenging in mice and rats. Common human T_1_ mapping methods with inversion recovery (IR) schemes [33,34,35], such as modified Look-Locker inversion recovery (MOLLI), shortened MOLLI (ShMOLLI), and saturation recovery single-shot acquisition (SASHA), usually require breath-holding and electrocardiogram (ECG) gating to eliminate cardiac and respiratory motion artifacts. Rodent CMR with respiratory and ECG gating can prolong scan time under anesthesia, which can compromise physiological and hemodynamic functions. Currently available MRI-compatible rodent ECG systems are not robustly shielded from magnetic field distortion. ECG waveforms often get distorted by the magnet, making accurate and effective ECG gating problematic. Additionally, in order to quantify myocardial T_1_ with and without Gd, it requires an intravenous (i.v.) line to administer Gd inside the magnet. Repetitive i.v. lines are challenging for rodents. Tail or jugular vein catheters are not MRI compatible and can be easily clotted or collapsed. Femoral i.v. lines are invasive and terminal, thus not feasible for longitudinal repetitive injections.

To address these challenges, the objective of this study is to establish a free-breathing, gating-free ECV procedure with a minimally invasive subcutaneous catheter for in-magnet Gd administration that can enable longitudinal repetitive ECV evaluations in rodent models. We leveraged the IntraGate sequence with a varied flip angle (VFA) for fast quantitative T_1_ mapping without needing ECG or respiratory gating [36,37,38,39]. Additionally, left ventricular (LV) blood is commonly used to normalize myocardial T_1_ values along with hematocrits to quantify ECV [15,16,17,18,19,20,21]. However, VFA-measured-T_1_ values in blood are altered by turbulent flow in-slice and especially through-slice flow, which introduces proton spins into the LV excitation region prior to the steady state. We have successfully overcome this issue and established a robust and reproducible ECV protocol in rodents with triple jugular vein T_1_ mapping using minimally invasive subcutaneous Gd administration.

## 2. Methods

### 2.1. Animals

Eight male Sprague Dawley (CD® IGS, Strain Code 001) rats were purchased from Charles River Laboratories. All animals received humane care in compliance with the NIH Office of Laboratory Animal Welfare (OLAW) guidelines. The animal protocol was approved by the Institutional Animal Care and Use Committee (IACUC) of the University of Pittsburgh (IACUC protocol number: 24034743; approval date: 25 March 2024; valid period: 25 March 2024–25 March 2027). Animals were housed in cages with a centralized filtered air/clean water supply system in a secure AALAS-certified vivarium in the Rangos Research Center of the Children’s Hospital of Pittsburgh of UPMC. Animal care is provided seven days a week based on the NIH *Guide for the Care and Use of Laboratory Animals.* Animals were kept in a 12:12 h dark/light cycle with ad libitum food and water.

All data was generated and analyzed in accordance with ARRIVE [40] guidelines, including protocols for blinding, randomization, counterbalancing, inclusion of proper controls, and appropriate statistical power.

To test the reproducibility of the method, each rat was repetitively imaged on 2 different days.

### 2.2. Anesthesia for Animal Imaging

All rats received general inhalation anesthesia with isoflurane for in vivo imaging. A rat was placed into a clear plexiglass anesthesia induction box that allowed unimpeded visual monitoring of the animal. Induction was achieved via administration of 2–3% isoflurane mixed with oxygen for a few minutes. Depth of anesthesia was monitored by toe reflex (extension of limbs, spine positioning) and respiration rates. Once the plane of anesthesia was established, the rat was maintained with 1–2% isoflurane in oxygen via a designated nose cone, and then the rat was transferred to the designated animal bed for MRI. The total anesthesia time is 45 min–30 min for imaging and 15 min for anesthesia induction, preparation, positioning, and post-imaging recovery. 

The core body temperature was monitored using a rectal optical probe (SA Instruments, Inc. Stony Brook, NY 11790, USA) and was maintained by circulating warm water inside the animal bed or a feedback-controlled MR-compatible air heater module for small animals (SA Instruments, Inc. Model 1025, Stony Brook, NY 11790, USA). Respiration was monitored with a pneumatic pillow sensor coupled to a pressure transducer. A total of *n* = 8 rats was included in the naïve cohort, while *n* = 3 were included in the ischemia–reperfusion injury cohort.

### 2.3. Subcutaneous Catheter for Gd Administration

A subcutaneous (subQ) catheter (McKesson Peripheral IV Catheter 25G × 0.75”, product number: 854664 Irving, Texas 75039, USA) was placed on the backside of each rat (Figure 1A) for anesthesia induction. The catheter was then connected to a microbore extension set (Braun Medical 36 in, reference number: V6203 Bethlehem, PA 18018, USA) to enable Gd administration while the animal was inside the Bruker BioSpec 70/30 USR spectrometer (Bruker BioSpin MRI, Billerica, MA, USA). 

A single bolus of clinical-grade Gadobenate Dimeglumine (MultiHance®, Bracco Diagnostics, Inc. Monroe Township, NJ 08831. USA NDC 0270-5164-12) with a dosage of 0.2 mmol/kg of bodyweight was injected via a subcutaneous catheter at time = 0 (Figure 2B). The specific method and dosage of gadolinium administration was optimized for our imaging procedure and processing pipeline.

### 2.4. Hematocrit

A blood sample was collected via facial vein using a 5.5 mm sterile blood lancet (Goldenrod Animal Lancet Medipoint, Inc Mineola, NY 11501, USA). The blood was sealed with clay in a micro-hematocrit capillary tube and spun in a micro-hematocrit centrifuge for 5 min to separate the blood cells and plasma. Hematocrit was then measured as a ratio of the final plasma volume to the total blood volume.

### 2.5. CMR Acquisition

All in vivo CMR was performed using a Bruker BioSpec 70/30 USR spectrometer (Bruker BioSpin MRI, Billerica, MA, USA) operating at a 7-Tesla field strength with a 35 mm quadrature volume coil for both transmission and reception. Fast free-breathing-no-gating CMR was acquired with IntraGate sequence, a fast low-angle shot (FLASH)-based pulse sequence with retrospective gating without needing ECG or respiratory triggering for acquisition [36,37]. Cardiac and respiratory motions were disentangled by the Bruker IntraGate interface for cardiac and respiratory cycles for retrospective imaging reconstruction. Acquisition performed for a single mid-slice of the mouse heart at the pupillary muscle level.

Dynamic contrast enhancement (DCE) time course: T_1_-weighted DCE time course (Figure 1B) after a single bolus Gd injection at time = 0 was acquired with the following parameters: Field of view (FOV) = 5 × 5 cm, slice thickness (SLTH) = 1.5 mm, sampled with an acquisition matrix of 256 × 256, in-plane resolution of 0.195 mm, echo time (TE) = 3.06 ms, repetition time (TR) = 5.65 ms, flip angle (FA) = 10°, number of repetitions (NR) = 3000, a total acquisition time (TT) = 36 min 12 s, and the frame rate = 0.092 frames per second (fps).

CMR acquisition protocol for ECV: Pre-contrast T_1_ mapping with VFA for the heart (Figure 2B red block 1) and jugular vein (Figure 2B blue block 1) was acquired before Gd bolus at time 0 (Figure 2B yellow block), followed by cine MRI (Figure 2B green block) for multi-slice short-axis (SA), long-axis 2-chamber (LA2C), and long-axis 4-chamber (LA4C) imaging. Once the single steady state was reached at around ~20 min post-Gd bolus, post-contrast T_1_ mapping for the heart (Figure 2B red block 2) was acquired with VFA flanked by 2 jugular vein T_1_ mapping (Figure 2B blue blocks 2 and 3) immediately before and after heart T_1_ mapping (Figure 2B red block 2).

Quantitative T_1_ mapping for the jugular vein (JV) blood: Fast free-breathing-no-gating T_1_ mapping for a jugular vein with 4 flip angles was acquired with the following parameters: FOV = 4.5 cm × 4.5 cm, matrix = 128 × 128, in-plane resolution = 0.352 mm, SLTH = 2 mm, FA = 3°, 19°, 22°, 28°, TE = 2.349 ms, TR = 77.2 ms, NR = 10, and TT = 49 s. The imaging plane was aligned with JV and completely included JV to avoid through-plane flow. The entire JV blood flow was in-slice with no through-slice flow.

Quantitative T_1_ mapping for the heart: Fast free-breathing-no-gating T_1_ mapping with 4 flip angles was acquired for the heart with the following parameters: FOV = 5 cm × 5 cm, matrix = 256 × 256, in-plane resolution = 0.195 mm, SLTH = 1.5 mm, FA = 3°, 19°, 22°, 28°, TE = 3.06 ms, TR = 200 ms, NR = 10, and TT = 2 min 26 s.

Cine CMR: Fast free-breathing-no-gating cine CMR for multi-slice short-axis (SA), long-axis 2-chamber (LA2C), and long-axis 4-chamber (LA4C) cine CMR was acquired with the following parameters: FOV = 5 cm × 5 cm, SLTH = 1.5 mm for SA and 2 mm for LA, matrix = 256 × 256, in-plane resolution = 0.195 mm, TE = 3.06 ms, TR = 5.65 ms, FA = 10°, NR = 250, cardiac phases = 20, and TT = 3 min 2 s each.

### 2.6. Systolic Cardiac Functions and Strain Analysis

Systolic cardiac functions were analyzed using Circle cvi42 Version 5.13.10 (Circle Cardiovascular Imaging, Inc. Calgary, Alberta, Canada) software, including LV volume, ejection fraction (EF), and stroke volume (SV). Strains are values that quantify the extent of ventricular deformation throughout cardiac phases: stretching/elongation or compression/shortening. Three classes of orthogonal principal strains were analyzed using Circle cvi42: circumferential, radial, and longitudinal strain for each cardiac phase.

### 2.7. Cardiac Phase Registration for T_1_ and ECV Quantification

To quantify T_1_ at the end-diastolic (ED) phase, singular value decomposition (SVD) was employed to automatically detect the end diastole (ED) phase from the IntraGate VFA time series. Each VFA time series was processed independently. First, the time series was reshaped as row vectors. Then, an average phase series was reconstructed by averaging the reconstruction of just the first eigenvalue of the left and right eigenvectors. The sign of this signal was taken, and the max value of the sign-time series was defined as ED.

### 2.8. T_1_ Mapping, ΔR_1_ Calculation, and ECV Mapping

Increased Gd uptake resulted in a brighter MR signal in T_1_-weighted images. The differences in overall T_1_ reduction are proportional to the amount of Gd taken up by the tissue. The longitudinal relaxation rate (*R*_1_ = 1/T_1_), the inverse of the longitudinal relaxation time (T_1_), was quantitatively mapped from the IntraGate VFA images using a variable projection approach combined with nonlinear least squares optimization [39,40]. We first computed the *R*_1_-dependent signal decay for the pre- and post-Gd administration inversion times. We model the MR signal’s nonlinear dependence on *R*_1_ via a combinatory equation that combines FA effects and *R*_1_ decay into the following equation:$$S\left(\alpha ;M_{z}^{0},R_{1}\right)=M_{z}^{0}\frac{1-e^{-T_{R}R_{1}}}{1-\cos\alpha e^{-T_{R}R_{1}}}\sin\alpha$$
where α represents the FA of the image, and $M_{z}^{0}$ represents the initial magnetization amplitude. The variable projection method was used to separate the linear ($M_{z}^{0}$ amplitude) and nonlinear (*R*_1_) parameters to reduce dimensionality during nonlinear fitting. The final *R*_1_ values are reshaped into an image with pre-Gd and post-Gd *R*_1_.

ECV quantification was performed using the following equation:$$\left(1-hematocrit\right)\frac{∆R_{1}\;Myocardium}{∆R_{1}\;Normalization\;Blood}$$

For repeatability experiments, each rat underwent duplicate imaging protocols. Hematocrit was measured once for each rat and used to calculate ECV for both scans. The $∆R_{1}\;Normalization\;Blood$ term is blood sourced from either LV blood pool or jugular vein blood. For every dataset, the left ventricle is manually segmented in ITK Snap (v3.6, ITK-Snap). All reported ECV and *R*_1_ values are the median value within the manually segmented LV region of interest (ROI).

### 2.9. Ischemia–Reperfusion Injury

Anesthesia was induced using 2.5% inhaled isoflurane in 2-month-old male rats weighing 275 ± 25 g. A total of *n* = 3 rats was utilized for initial validation exploration. The rat was put on RoVent® Jr. small animal ventilator (Kent Scientific Corporation, Torrington, CT, USA). The chest of the rat was shaved. Thoracotomy was performed; the thoracic cavity was opened between 4th and 5th ribs of the left side of the rat. A surgical retractor was placed between the ribs to keep the cavity open. The pericardium was cleared to make the heart accessible for left anterior descending (LAD) coronary artery ligation. By gently pushing the right-side ribs, the heart was temporarily brought out of the chest near the cavity opening, and the LAD coronary artery was located and ligated using a 6-0 surgical suture to create ischemia. A flexible rubber tubing was placed between the suture knots to safely enable the untying of the knot after the designated ligation time without injuring the myocardium. The left ventricle turned pale distal to the ligation site immediately after inducing ischemia. After LAD ligation, the heart was slid back into its position, and the cavity was temporarily closed. After 60 min, the ligation was released, and the thoracic cavity was closed. The rats were recovered from anesthesia and housed in individual cages for 28 days to allow for the development of myocardial fibrosis.

### 2.10. Statistical Analysis

Descriptive statistics were used for normally distributed variables. A combination of the D’Agostino and Pearson and the Shapiro–Wilk test was used to assess the normality of datasets. If the dataset failed normality, the median and interquartile range (IQR) were utilized to describe the distribution. A two-tailed paired t-test was used with α = 0.05 for normally distributed data. For non-normally distributed data, the Wilcoxon matched-pairs signed-rank test was used. All statistical analysis was performed in Prism (v10.2.3, GraphPad Boston, MA, USA). Summary of all normality testing results can be found in the Appendix A.

## 3. Results

### 3.1. Free-Breathing, Gating-Free Cine, and Dynamic Contrast Enhancement (DCE) Time Series with a Single Bolus Subcutaneous (subQ) Gd Administration

A single bolus of Gd can be successfully administered to an anesthetized rat inside the MRI scanner via a sterile subcutaneous (subQ) catheter (Figure 1A). This enables pre- and post-contrast acquisition without moving the animal. As subQ catheters are minimally invasive, this enables repetitive imaging using the same animals. Fast free-breathing, gating-free cine (Figure 1B) and dynamic contrast enhancement (DCE, Figure 1C) time series can be acquired with IntraGate, (Paravision 5.1 Bruker BioSpin MRI, Billerica, MA, USA) which convolves cardiac and respiratory motions for image reconstruction. Figure 1B shows 8 out of the 20 cardiac phases for a mid-SA slice at the pupillary muscle level. A DCE time course (Figure 1C) can be successfully reconstructed into a single cardiac phase (ED, end diastole), allowing visualization of the contrast-specific dynamics over time without motion artifacts.

### 3.2. Acquisition Schema and DCE Time Courses

The dynamic contrast enhancement (DCE) signal time course was utilized to understand when the signal contrast reaches a steady state to allow for consistent, repeatable acquisition of pre- and post-contrast images of the heart and jugular vein. LV myocardium (Figure 2A) after a single subQ Gd bolus reached a steady state at around ~20 min. Fast T_1_ mapping can be successfully acquired with IntraGate without ECG nor respiratory gating with four FAs (3°, 19°, 22°, 28°) for jugular vein blood (49 s, Figure 2B blue blocks) and the heart (2 min 26 s, Figure 2B red blocks) before and after Gd bolus (Figure 2B, yellow block). It is important to acquire post-Gd contrast T_1_ mapping ~20 min after the Gd bolus during the steady state. To prevent potential quasi-steady-state fluctuation, the post-contrast jugular vein blood T_1_ mapping was repeated twice—once before (Figure 2B blue block 2) and once after (Figure 2B blue block 3) heart T_1_ mapping (Figure 2B red block 2). The two post-contrast jugular vein blood T_1_ were averaged to calculate myocardial ECV. During the waiting period for the heart signals to reach the steady state after the Gd bolus, the full cine CMR (Figure 2B green block) with 20 cardiac phases was acquired for multi-slice short-axis (SA), long-axis two-chamber (LA2C), and long-axis four-chamber (LA4C). The cine CMR was acquired for global systolic function and strain analysis, including LV volumes, ejection fractions, stroke volumes, and three orthogonal principal strains (circumferential, longitudinal, and radial).

The mean DCE time course of LV myocardium (Figure 2C) and LV blood pool (Figure 2D) across all eight rats demonstrated similar dynamics with a steady increase until 20 min, when both reached the steady state. The overall variance across the time courses for the LV blood pool (STD LV blood pool 0.11) is higher than that of the LV myocardium (STD LV myocardium 0.078). The mean DCE time course for the jugular vein (Figure 2E) of the eight animals showed a more rapid, logarithmic increase in the steady state by 15 min post-Gd administration and was held steady past 30 min. The overall variance of the JV (STD JV 0.089) is lower than that of the LV blood pool (STD LV blood pool 0.11).

### 3.3. T_1_ Mapping with Varied Flip Angles (VFA)

Free-breathing gating-free T_1_ mapping was achieved using voxel-wise T_1_ calculation with four flip angles (FA, 3°, 19°, 22°, and 28°). Figure 3 shows a mid-SA slice (Figure 3A) at the pupillary muscle level and jugular vein (Figure 3B) with different FAs before (top row) and after (bottom row) Gd contrast. We calculated the average native MR signal intensity in the LV myocardium and LV blood pool (Figure 3A) and the jugular vein (Figure 3B) for each FA. We plotted the MR intensity as a function of the flip angle (3°, 19°, 22°, and 28°) and observed a logarithmic response in both the LV myocardium (Figure 3C) and the LV blood pool (Figure 3D). The final 28° flip angle images showed a 1.64 and 1.28-fold increase in contrast for the LV myocardium and LV blood pool, respectively. In the jugular vein (Figure 3E), we observed a more linear increase in MR intensity as a function of the VFA, with a final 1.17-fold contrast gain.

We performed quantitative *R*_1_ mapping of the LV myocardium (Figure 4) pre- and post-Gd administration for eight animals, scanned in duplicates. Figure 4A shows an example of LV myocardial *R*_1_ maps pre- (Figure 4A left) and post- (Figure 4A middle) Gd contrast, as well as the Δ*R*_1_ map (Figure 4A right), showing the differences between the two, for the same animal with two separate scans. The mean pre-contrast *R*_1_ in Sprague Dawley rat myocardial tissue was 2.31 ± 1.16 s^−1^ (Figure 4B). The post-contrast quantitative *R*_1_ across the same scans was 3.47 ± 1.93 s^−1^ (Figure 4C). The overall mean Δ*R*_1_ for LV myocardial tissue was 1.12 ± 0.80 s^−1^ (Figure 4D).

We performed quantitative *R*_1_ mapping of the blood pools (Figure 5) pre- and post-Gd administration for eight animals, scanned in duplicates. Figure 5A shows an example of LV blood (Figure 5A) and jugular vein (Figure 5B) *R*_1_ maps pre- (Figure 5A,B left) and post- (Figure 5A,B middle) Gd contrast, as well as the Δ*R*_1_ map (Figure 5A,B right), as the differences between the two for the same animal with two different scans. The LV blood (Figure 5A) showed higher variations in *R*_1_ and Δ*R*_1_ due to inflow turbulence and blood mixing in the LV. Across the sixteen scans, we quantified the median Δ*R*_1_ for LV blood (Figure 5B) and jugular vein blood (Figure 5D). The mean Δ*R*_1_ for the LV blood pool (Figure 5B) was found to be 13.42 ± 7.15 s^−1^, whereas the mean Δ*R*_1_ for the jugular vein was 1.92 ± 0.49 s^−1^. Due to differences in the magnitude of the mean across the three datasets, the coefficient of variation was calculated to provide an unbiased measure of variability. The coefficient of variation for the LV blood pool and JV was 53.2% and 25.3%, respectively. LV blood pools had much larger variations in *R*_1_ and Δ*R*_1_ due to inflow turbulence and blood mixing in the LV. Jugular vein blood had more consistent *R*_1_ and Δ*R*_1_.

### 3.4. ECV Quantification and Reproducibility

The subcutaneous catheter is minimally invasive and can enable repetitive ECV quantifications. Each rat was scanned twice on two different days to test the reproducibility of the ECV quantification. Myocardial ECV was calculated twice using either LV blood (Figure 6A,B) or jugular vein blood (Figure 6C,D) for blood pool normalization. Hematocrit was taken once for all eight animals; the same hematocrit value for a given animal was used to evaluate all ECV maps. The average hematocrit for the eight animals was found to be 47.4 ± 1.64%.

First, we evaluated LV myocardial ECV using LV blood normalization (Figure 6A,B), following human myocardial ECV convention. Qualitatively, there were few differences between replicate 1 (Figure 6A top row) and replicate 2 (Figure 6A bottom row) of the same animals. We calculated the median ECV of each scan performed and evaluated differences between the means of scan 1 and scan 2 ECV distributions. We found no difference between scan 1 (Mean ECV: 0.048 ± 0.018) and scan 2 (Mean ECV: 0.041 ± 0.017) using paired t-testing between the two groups (*p* = 0.60). We then evaluated ECV repeatability using jugular vein blood for normalization (Figure 6C,D). Again, qualitatively minimal differences were found from scan to scan (Figure 6D). The scan 1 and scan 2 distributions had means of 0.249 ± 0.072% and 0.249 ± 0.068, respectively (Figure 6D). We found no difference between scan 1 and scan 2 using paired t-testing between the two groups (*p* = 0.99). The repetitive ECV quantifications in the same animals on different days showed excellent reproducibility.

The ECV quantification by LV blood and or jugular vein blood normalization differed by two main factors: the magnitude of ECV and within-distribution variation for each set of duplicate scans. The magnitude of ECV was ~5-fold higher using jugular vein blood versus LV blood for normalization. The mean myocardial ECV (0.048) obtained by LV blood normalization was non-physiological, whereas the mean myocardial ECV (0.25) obtained by jugular vein blood normalization was comparable to literature reports of baseline myocardial ECV in both humans [15,16,17,18,19,20,21] and rodents [23,41,42,43,44]. Additionally, the coefficient of variation across the 16 scans when utilizing the LV blood pool for normalization was 39.4%, whereas when using jugular vein blood it was 28.1%. The within-subject variability, that is, the average difference between scans for a given subject, was 25.5% for jugular vein normalization versus 44.3% for LV normalization.

### 3.5. ECV Quantification for Fibrosis After Ischemia–Reperfusion Injury

We tested whether this ECV protocol can be applied in the pathological condition of ischemia–reperfusion injury (IRI) model (Figure 7). A total of 28 days after IRI, ECV near the IRI site was elevated (Figure 7A, yellow arrow), indicating increased myocardial fibrosis. This was validated by histological trichrome staining (Figure 7B, yellow arrow) showing blue collagen deposits in the corresponding region of the myocardium. Strain analysis (Figure 7C–F) showed compromised strain in the corresponding region of the myocardium. Our data showed that this ECV protocol can be used to detect pathological conditions.

## 4. Discussion

### 4.1. Minimally Invasive Subcutaneous Protocol for ECV Quantification

We successfully established a minimally invasive, fast ECV quantification protocol to enable longitudinal repetitive ECV quantifications in rodents. This is the first work, to our knowledge, to both optimize a subQ Gd administration procedure combined with VFA imaging and repetitive scans to quantify ECV with demonstrated repeatability in rodents. Our protocol uses a subcutaneous catheter to deliver a Gd bolus inside the MRI scanner. Our data demonstrated that subcutaneous Gd bolus administration induced a reasonable DCE time course, reaching a steady state for stable T_1_ (or *R*_1_) quantification. Furthermore, our protocol uses a free-breathing, gating-free variable flip angle (VFA) T_1_ (or *R*_1_) quantification scheme leveraging the IntraGate sequence with retrospective gating. This allows for rapid acquisition of cardiac (~2.5 min) and jugular vein (49 s) T_1_ (or *R*_1_) quantification with no cardiac nor respiratory motion artifacts. In addition, full cine CMR can be acquired during the post-Gd waiting period using IntraGate for full functional analysis. The entire full protocol for pre- and post-contrast T_1_ (or *R*_1_) quantification with full cine CMR can be completed within 30 min without ECG or respiratory gating in anesthetized rodents. This protocol is simple, robust, and highly reproducible. Our data demonstrated excellent reproducibility between two repeated scans on the same animals performed on different days. The minimally invasive subcutaneous catheter protocol can be repeated many times. This simple, minimally invasive protocol can facilitate longitudinal monitoring of ECV with full cardiac function for disease progression. It can be used to monitor therapeutic interventions for disease regression in rodent models.

### 4.2. Jugular Vein vs. LV Blood Pool Normalization

Conventional ECV protocols [15,16,17,18,19,20,21,22] use inversion recovery schemes for T_1_ (or *R*_1_) quantification [33,34,35], such as modified Look-Locker inversion recovery (MOLLI), shortened MOLLI (ShMOLLI), and saturation recovery single-shot acquisition (SASHA). The inversion pulses remove proton spins on the acquisition planes, including the incoming blood to the LV. There are usually little residual fresh spins on the acquisition planes. With the VFA method, the fresh proton spins in the incoming blood can introduce variable blood signal intensities due to inflow turbulence and fresh blood mixing in the LV. As a result, ECV normalization using the LV blood pool can be variable and inaccurate. In addition, fresh inflow proton spins can make blood T_1_ appear to be different, resulting in abnormally low ECV values (0.048). One way to overcome this inflow artifact is to implement a crusher gradient or a long, low-power adiabatic pulse to saturate the incoming fresh spins before T_1_ (or *R*_1_) acquisition. However, an additional crusher gradient or adiabatic pulse can prolong the scan time, which can defeat the purpose of fast acquisition. In addition, inflows for the heart can come from variable directions from the pulmonary vein, superior vena, and inferior vena cava. The inflows from the left atrium to the left ventricle through the mitral valve can be turbulent. These can make the complete saturation of incoming spins in inflows variable.

Therefore, we implemented an alternative jugular vein blood pool for normalization. The JV imaging plane was aligned with JV blood flow and included the entire JV completely, thus JV blood flow was all in-slice without through-plane flow. To avoid potential quasi-steady-state fluctuation of Gd in the blood, we acquired two post-contrast jugular vein T_1_ (or *R*_1_) quantifications immediately flanking the heart T_1_ (or *R*_1_) acquisition, then used the average of the two for ECV quantification. Our data demonstrated that LV myocardial ECV quantification using the triple jugular vein T_1_ (or *R*_1_) acquisitions (one pre-contrast and two post-contrast) is highly reproducible between repeated scans on the same animals. Furthermore, the mean baseline ECV value (0.249) obtained for the Sprague Dawley rats using triple jugular vein blood T_1_ (or *R*_1_) normalization was physiological and comparable to the reported baseline ECV values in both humans [15,16,17,18,19,20,21] and rodents [23,40,41,42].

### 4.3. Future Work and Additional Considerations

Our singular value decomposition algorithm for robust automated strain estimation has not been evaluated under extreme cardio-pathologic conditions, such as advanced heart failure or in cases of extreme subject motion. To demonstrate more robust potential for clinical translation, we will explore the efficacy of the proposed algorithm for cardiac phase detection under more extreme circumstances, such as cardiac models with deleterious cardiac motion and arbitrary motion artifacts, such as respiratory or subject motion.

The work proposed does not account for or consider ECV quantification errors due to inhomogeneity in the B1 field. The same experimental setup described here will be repeated in the future, but B1 field mapping will be performed to understand B1 inhomogeneity and its effect on errors in ECV quantification. Alternative MR imaging sequences, such as SASHA [45], may be evaluated if it is determined that B1-inhomogeneity-specific ECV measurement errors are prevalent in the proposed schema.

### 4.4. Mouse Specific Considerations

Mice present a number of physiological differences compared to rats that must be addressed for successful translation of the procedure. Specifically, mice are overall much smaller than rats, with smaller vessels, cardiac structures, and higher heart rates. Smaller vasculature structures and cardiac structures increase the risk of through-plane flow artifacts and partial volume effects that blur vessel borders and disrupt T1-mapping accuracy. Overcoming these challenges will require higher overall resolution and slightly longer scan times. The Bruker IntraGate interface has been developed on both mice and rats for pre-clinical imaging, and it is not expected that the increased heart rate will hurt image quality or introduce motion artifacts.

### 4.5. Translation Considerations

Late Gadolinium Enhancement (LGE) and T1-mapping with ECV quantification are currently utilized to measure fibrotic deposition and interstitial fibrosis in humans [35,46,47], respectively. If the approach demonstrated here was adapted for human use, this approach could facilitate more robust, fast, and repeatable quantification of diffuse myocardial fibrosis, especially in settings where minimizing scan time, motion artifacts, and invasive procedures is critical. The jugular vein normalization method may also improve the accuracy of ECV quantification compared to conventional LV blood pool techniques, potentially advancing precision cardiac imaging and increasing reliability in clinical trials and longitudinal patient monitoring.

## 5. Limitations

This fast, minimally invasive ECV protocol requires the steady state of Gd in the blood and myocardium. This protocol cannot work if the steady state cannot be achieved when the Gd dosage is too low, such that the subcutaneous Gd absorption is too slow and becomes the rate-limiting step. Thus, an optimal Gd dosage is needed for this protocol.

Using jugular vein blood for ECV quantification can mitigate inflow artifacts due to incoming fresh proton spins draining from the great veins into the heart. However, this cannot eliminate the fresh proton spins partitioning into the myocardium via the coronary arteries. Thus, this protocol will slightly overestimate the myocardial T_1_.

In order to measure blood T_1_ (or *R*_1_) in jugular veins as closely as possible to the time of myocardial T_1_ (or *R*_1_) acquisition, we used a very short jugular vein VTR protocol with only 49 s for all 4 FAs. This was accomplished using lower spatial resolutions. The drawback of this approach is the potential partial volume artifact of the jugular veins. To prevent the partial volume artifact of jugular veins, the voxel sizes of the jugular vein T_1_ (or *R*_1_) acquisitions need to be optimized based on the size of the jugular veins, in particular for mice or juvenile animals.

Physiological changes due to anesthesia are an important consideration in any preclinical animal imaging study. ECV is a measure of cardiac physiological status and may be affected by anesthetic-induced changes. However, as ECV is a measure of interstitial fibrosis—a physiological parameter that is not readily altered under rapid temporal dynamics, our 30 min scan time, combined with correlation across similar studies evaluating rodent ECV [23,41,42,43,44]—it is not expected to cause a significant effect in ECV measurement.

## 6. Conclusions

We established a simple, minimally invasive, fast, and robust CMR protocol in rodents that can enable longitudinal repetitive ECV quantifications for cardiovascular disease progression and regression with interventions. Maybe elaborate more here, although I don’t know what precedent you use for templating your cocnclusion.

## Figures and Tables

**Figure 1 biomolecules-15-01732-f001:**
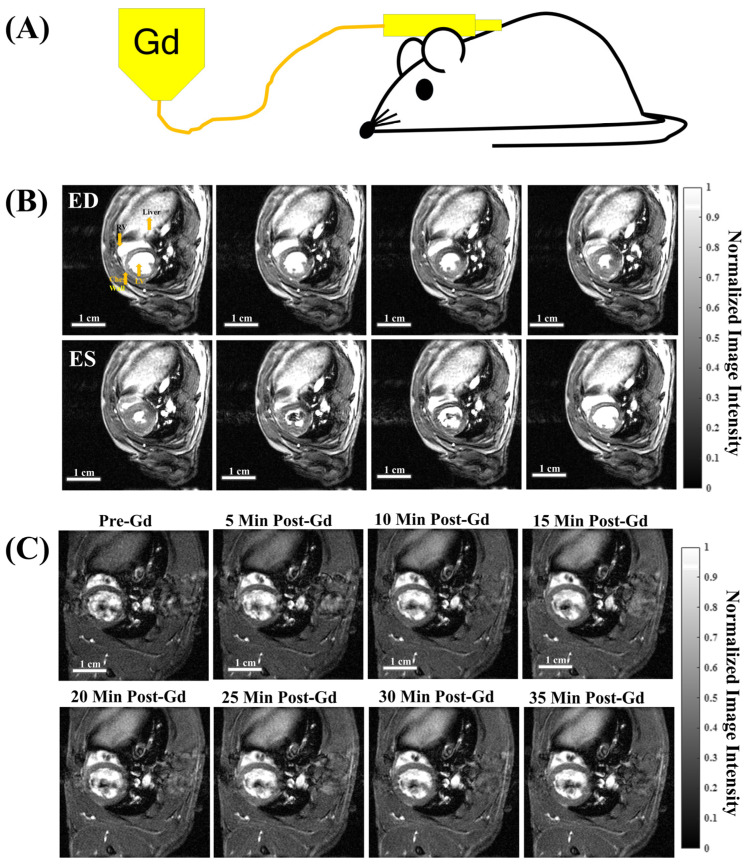
Free-breathing, gating-free cine and dynamic contrast enhancement (DCE) time series with subQ Gd administration. (**A**) A single bolus of Gd was administered via a subcutaneous (subQ) catheter on the back of the animal inside the scanner to enable pre- and post-contrast acquisitions. (**B**) Cine time series of a full cardiac cycle acquired by IntraGate with FA = 28°, showing 8 out of the 20 cardiac phases. ED = end diastole. ES = end systole. (**C**) Dynamic contrast enhancement (DCE) time series at ED phase showing the overall increase in contrast from the time of Gd bolus injection. the Gd bolus was 0.2 mmol/Kg of bodyweight, and the total acquisition time was 36 min and 12 s.

**Figure 2 biomolecules-15-01732-f002:**
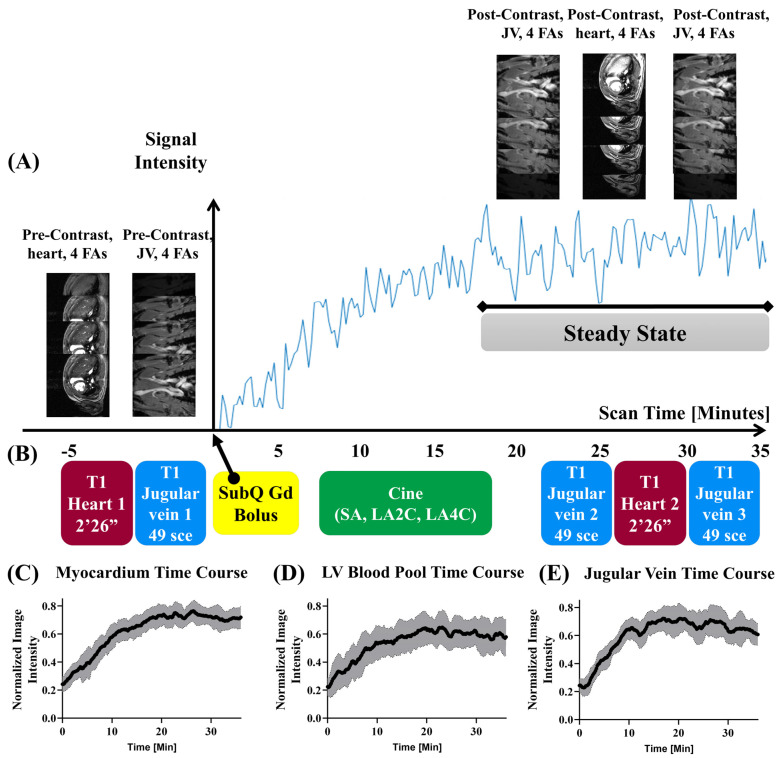
Image acquisition schema and DCE time course. (**A**) Temporal LV myocardial signal profile with dynamic contrast enhancement (DCE) for a single, representative animal. *Y*-axis: signal intensity of LV myocardium. *X*-axis: time (minutes). Gd administration occurs at t = 0 (yellow box). The LV myocardial signals reached steady state around ~20 min post-Gd bolus. (**B**) Imaging acquisition scheme. Yellow blocks: Gd bolus of 0.2 mmol/Kg of bodyweight. Red blocks: T_1_ mapping for the heart. Blue blocks: T_1_ mapping for jugular veins. Green block: cine CMR for multi-slice SA, LA2C, and LA4C. Prior to Gd injection, VFA heart and jugular vein image series were acquired at four different flip angles (FA): 3°, 19°, 22°, and 28°. After Gd injection, it is critical to wait 20 min to acquire the next set of post-Gd image sets to allow for steady-state contrast. When the steady state contrast was achieved, next sets of images with 4 FAs were acquired in the following order: jugular vein, heart, and jugular vein again. Two sets of jugular vein images were averaged together for blood T_1_ quantification. (**C**–**E**) Time courses of the dynamic contrast time series taken from (**C**) LV myocardium, (**D**) LV blood pool, and (**E**) jugular vein. All tissue and blood sources reached the steady state by 20 min. *Y*-axis is normalized image intensity. Black lines represent average signal intensity of all rats (*n* = 8), and shaded gray regions represent standard deviation of each time point.

**Figure 3 biomolecules-15-01732-f003:**
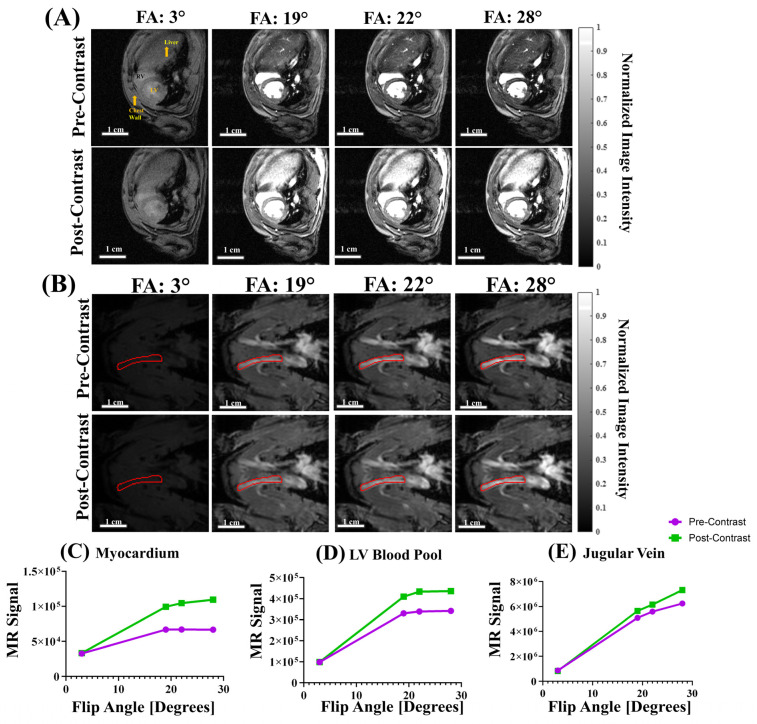
Varied flip angle (VFA) imaging of cardiac structures and the jugular vein. (**A**) ED images for each FA for both pre- (**top row**) and post- (**bottom row**) Gd administration. All images were normalized to the max image intensity of the post-contrast FA = 28° scan to show relative changes in image intensity. (**B**) Jugular vein VFA images for each FA used in this study pre- (**top row**) and post- (**bottom row**) contrast. The right jugular vein is outlined in red in each image. All images were normalized to the max image intensity of the post-contrast FA 28° image to show relative changes in image intensity. The red lines outline a jugular vein. (**C**–**E**) Signal intensity as a function of FA for both pre-Gd (purple) and post-Gd (green) series for (**C**) LV myocardium, (**D**) LV blood pool, and (**E**) jugular vein at 4 FA—3°, 19°, 22°, and 28°—of a single, representative rat. We observed a consistent logarithmic response in image intensity as a function of FA.

**Figure 4 biomolecules-15-01732-f004:**
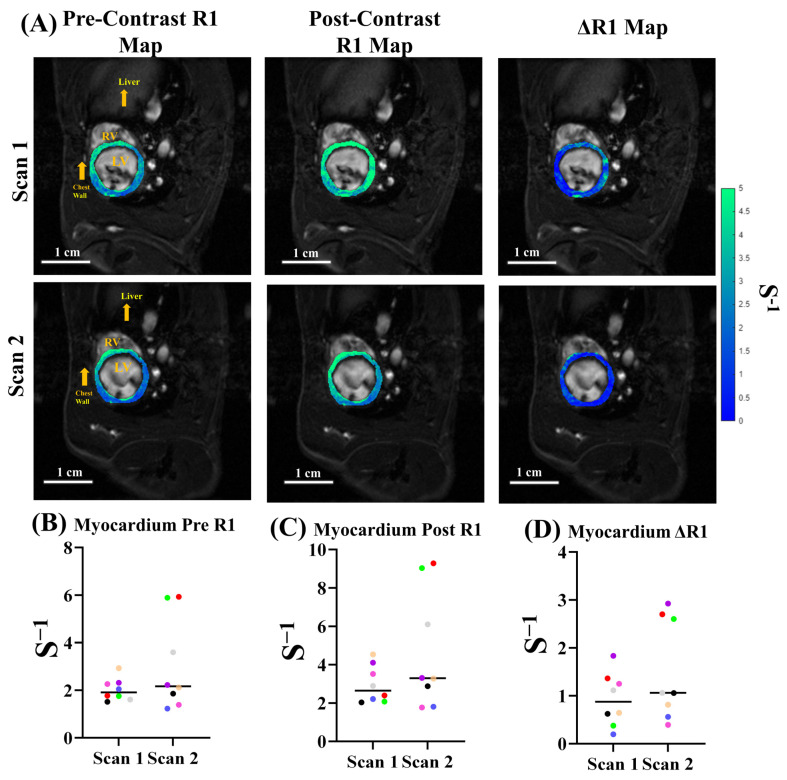
*R*_1_ mapping of LV myocardium. (**A**) *R*_1_ maps overlaid anatomical FA = 28° images at ED. Left: pre-contrast *R*_1_ map. Middle: post-Gd contrast *R*_1_ map. Right: Δ*R*_1_ map, the differences between pre- and post-Gd contrast. (**B**–**D**) Median *R*_1_ plots from *n* = 16 scans for (**B**) pre-contrast, (**C**) post-Gd contrast, and (**D**) Δ*R*_1_ of the LV myocardium for *n* = 16 scans. Dots of the same color come from the same animal for the two different scans.

**Figure 5 biomolecules-15-01732-f005:**
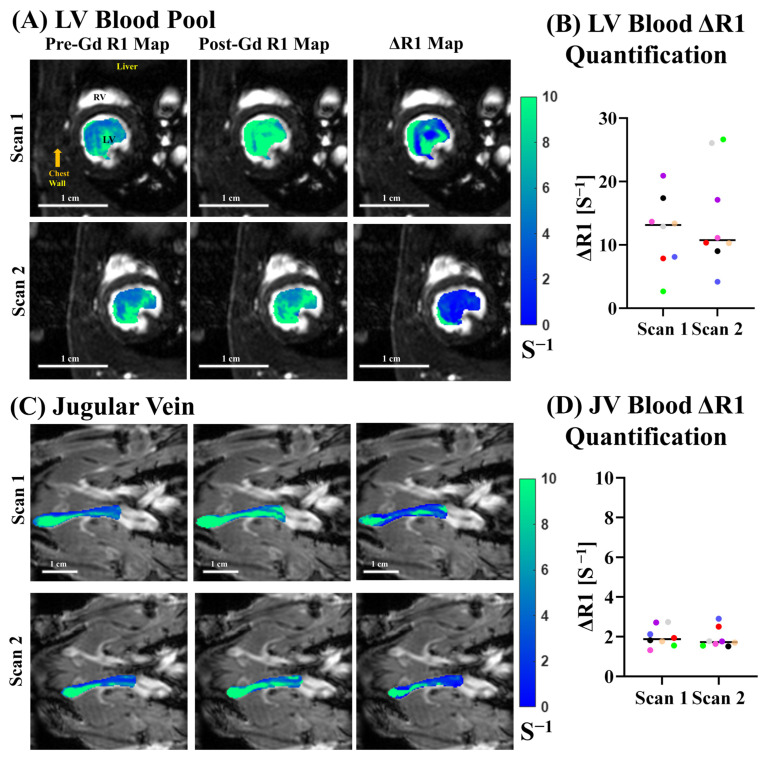
*R*_1_ mapping of blood pools. (**A**) *R*_1_ maps overlaid anatomical FA = 28° images at ED for the LV blood pool showing pre-Gd *R*_1_ maps (**left column**), post-Gd *R*_1_ maps (**middle column**), and Δ*R*_1_ maps (**right column**) for two scans from the same animal. (**B**) Median Δ*R*_1_ quantification for LV blood pool normalization sources for *n*-16 scans. Matching colors represent the same animal for each scan. (**C**) *R*_1_ maps overlaid anatomical FA = 28° images at ED for the jugular vein blood pool showing pre-Gd *R*_1_ maps (**left column**), post-Gd *R*_1_ maps (**middle column**), and Δ*R*_1_ maps (**right column**) for two scans from the same animal. (**D**) Median Δ*R*_1_ quantification for the jugular vein for *n* = 16 scans. Matching colors represent the same animal across two scans.

**Figure 6 biomolecules-15-01732-f006:**
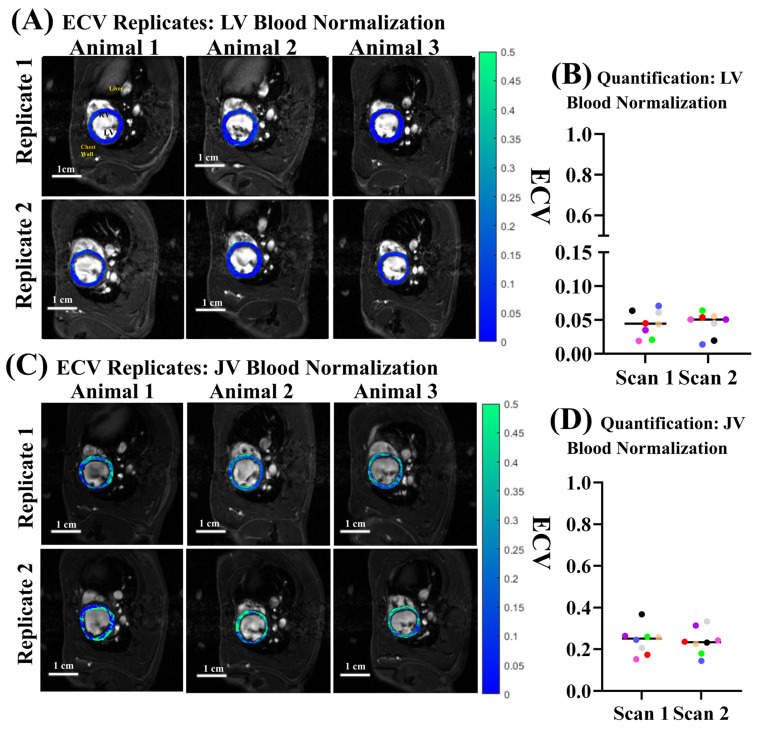
LV myocardial ECV repeatability from different blood sources for normalization. Each rat was imaged on 2 different days to quantify ECV. ECV was calculated twice with normalization by the LV blood pool (**A**,**B**) or jugular vein blood (**C**,**D**). (**A**) LV myocardial ECV maps of 3 rats on day 1 (**top row**) or day 2 (**bottom row**) using LV blood pool for normalization. LV myocardial ECV maps were overlaid on anatomical FA = 28° images at ED. (**B**) Median ECV for LV myocardium calculated twice with normalization by the LV blood pool on 2 different days. (**C**) LV myocardial ECV maps of 3 rats on day 1 (**top row**) or day 2 (**bottom row**) using jugular vein blood pool for normalization. LV myocardial ECV maps were overlaid on anatomical FA = 28° images at ED. (**D**) Median ECV of LV myocardium but with normalization via jugular vein blood pool on 2 different days. Dots of the same color come from the same animal for the two different scans.

**Figure 7 biomolecules-15-01732-f007:**
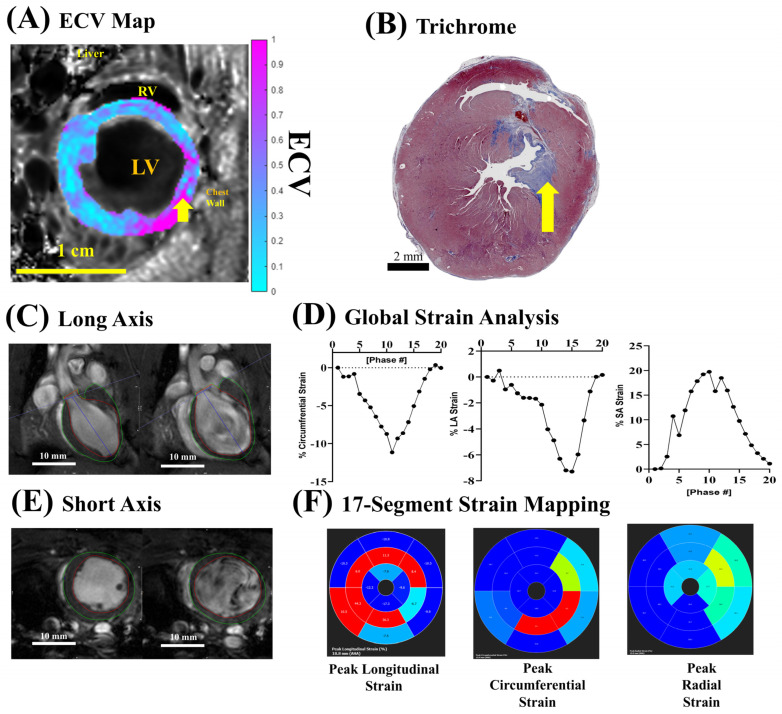
ECV mapping with ischemia–reperfusion injury (IRI). (**A**) LV myocardial ECV map overlaid on anatomical FA = 28° images at ED using jugular vein blood pool for normalization on day 28 after IRI. The yellow arrow points to the site with high ECV reflecting high fibrosis. (**B**) Trichrome histology of the same animal, taken at 40× magnification; scale bar is 2 mm. The yellow arrow points to the site with fibrotic deposition (blue), corresponding to high ECV regions in A. (**C**) Long-axis cine showing ES (**left**) and ED (**right**). (**D**) Short axis cine showing ES (**left**) and ED (**right**). (**E**) Global strain plots showing the % change in cardiac strain throughout the cardiac cycle for longitudinal strain (**left**), circumferential strain (**middle plot**), and radial strain (**right plot**). (**F**) Seventeen-segment maps showing the spatial strain distribution throughout the different regions of the rat heart. All data shown is for a single, representative IRI rat. A total of *n* = 3 IRI rats were imaged.

## Data Availability

The original contributions presented in this study are included in the article/Appendix A. Further inquiries can be directed to the corresponding author.

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
