# Peer review of "Minimally Invasive Free-Breathing Gating-Free Extracellular Cellular Volume Quantification for Repetitive Myocardial Fibrosis Evaluation in Rodents"

_biomolecules, 2025, doi:10.3390/biom15121732_

Round 1
Reviewer 1 Report
Comments and Suggestions for Authors
The aim of the study was to establish ECV quantification protocol using MRI in mice.
The introduction provides sufficient information about the topic to be addressed. It addresses the strengths and limitations of using rodent models in research, as well as the adaptation of MRI examination to rodents.
The study design is well explained, allowing the experiment to be reproduced. The results support the discussions. The study's limitations are also addressed. Overall, it is a good article with a robust and clearly presented design.
Anyway, a few minor suggestions:
1. How these results may influence the approach to myocardial fibrosis in humans.
2. The discussion section reproduces the results; I suggest comparing the results with other studies and highlighting the new elements introduced here
- If this is the first study to address this issue, it should be emphasized.
- Do you think the results could be influenced by the species of rodent used?
Author Response
Reviewer 1 Comments & Responses:
We thank the reviewer for the valuable comments and expert suggestions. We have made revisions according to the reviewer’s guidance, and believe this has greatly strengthened our manuscript. We answer the reviewer’s comments as below:
- How these results may influence the approach to myocardial fibrosis in humans. An additional section of the discussion has been added to address considerations in human translation of our work.
- The discussion section reproduces the results; I suggest comparing the results with other studies and highlighting the new elements introduced here. The overall discussion section has been revised with additional sections and comparisons added. Specific comparisons are made within section 4.2 beginning and end, end of section 4.3, throughout section 4.5 and in the limitations, section 5.
- If this is the first study to address this issue, it should be emphasized. Thank you for pointing this out. The following clarification has been added to section 4.1: “This is the first work, to our knowledge, to both optimize a subQ Gd aministration proedure combined with VFA imaging and repetitive scans to quantify ECV with demonstrated repeatability in rodents.”
- Do you think the results could be influenced by the species of rodent used? An additional section (4.4) has been added to address important considerations if applying this methodology to mice.
Reviewer 2 Report
Comments and Suggestions for Authors
Summary:
This is an animal study establishing in rats a novel and versatile ECV quantification protocol that is free-breathing and gating-free for quantitative cardiac MRI myocardial fibrosis evaluation. The method also poses minimal invasiveness that addresses past challenges of similar approaches for ECV measurement in rats. The endeavor demonstrated reproducible ECV quantification between repeated scans and successful application in ischemic reperfusion injury rats, detecting elevated ECV corresponding to histology-validated myocardial fibrosis. The objectives are clearly stated, and the study's rationales, synthesized context, and results meet its set objectives.
It is essential that core publishing ethics be upheld by methodology and data transparency, fair authorship attribution, and full disclosure of any conflicts of interest. Publishable data should suffice the stated research objectives and be free from any signs of manipulation. Methods should be written in a reproducible manner. On the reviewer's part, respectful conduct throughout the review process will be maintained for trust and integrity.
Comment:
- "Non-invasive subcutaneous catheter" is misleading, as any puncture of skin is by definition "invasive". Consider "less-invasive", "low-invasive", or "minimally invasive".
- The equations applied are standard in CMR; however, providing citations to peer literature would enhance their foundations.
- Whether the catheter was only attached to the animal after anesthesia was not stated; assuming this was the case, the total anesthesia duration was not stated for the entirety of ECV assessments. For adherence to ARRIVE (as cited), clarify these points to ensure reproducibility, particularly for future longitudinal studies.
- Clarify whether "a dosage of 0.2 mmol/kg bodyweight was injected via a subcutaneous catheter at time = 0" is standardized. Consider citations.
- The sample size for the ischemic reperfusion injury (IRI) validation was not clear (or is lacking). For transparency, the following numbers should be clear: # rats used in the IRI cohort, total # animals used in the study, and whether a control group was included.
- Figure 1's caption is missing the Gd dosage and total acquisition time (TT) for the DCE time series.
- Figure 2. The interchangeable use of "LA4D" (Long‑Axis 4‑Dimensional) vs. "LA4C" (Long‑Axis 4‑Chamber) in the main text appears misleading.
- Figure 2. 2A, unlike 2C-2E, does not specify whether the data represents a single animal or an average.
- Figure 3. 3C-E displays average signal intensity; hence, the sample size should be stated in the caption.
- Figure 6. (B) description is stated twice. The second instance should be the description for (D).
- Figure 7. 7B mentions capturing the image at 40x mag, but it is "shown at 0.8x magnification". This is ostensibly unclear and should be clarified. Is this display size reduction? Also, for "longitudinal strain (left), circumferential strain (middle plot), and radial strain (left plot)", should it be "radial strain (right plot)?
Author Response
Reviewer 2 Comments & Responses
We thank the reviewer for the valuable comments and expert suggestions. We have made revisions according to the reviewer’s guidance, and believe this has greatly strengthened our manuscript. We answer the reviewer’s comments as below:
- "Non-invasive subcutaneous catheter" is misleading, as any puncture of skin is by definition "invasive". Consider "less-invasive", "low-invasive", or "minimally invasive". Thank you for this clarification, the title and overall language throughout the paper has been revised with “minimally invasive” rather than non-invasive.
- The equations applied are standard in CMR; however, providing citations to peer literature would enhance their foundations. Additional citations have been added to the CMR equations to represent current existing literature. Specifically, citation 23 starting in the introduction and citations 39 and 40 of section 2.8.
- Whether the catheter was only attached to the animal after anesthesia was not stated; assuming this was the case, the total anesthesia duration was not stated for the entirety of ECV assessments. For adherence to ARRIVE (as cited), clarify these points to ensure reproducibility, particularly for future longitudinal studies. Thank for your pointing out this. We agree these details need clarified in order to optimize reproducibility of our work. The catheter was indeed attached after anesthesia induction, and the total anesthesia time of roughly 45 minutes was added to the methods section. This anesthesia time covers rodent setup and imaging – 30 min for imaging whereas 15 min for preparation, positioning, and post-imaging recovery.
- Clarify whether "a dosage of 0.2 mmol/kg bodyweight was injected via a subcutaneous catheter at time = 0" is standardized. Consider citations. Additional clarification was added concerning dosage of Gd administration. Gd was injected at time t=0 for all animals in the study.
- The sample size for the ischemic reperfusion injury (IRI) validation was not clear (or is lacking). For transparency, the following numbers should be clear: # rats used in the IRI cohort, total # animals used in the study, and whether a control group was included. Thank for your identifying these missing details that are crucial for scientific clarity. All animal cohort sizes are included throughout the revision.
- Figure 1's caption is missing the Gd dosage and total acquisition time (TT) for the DCE time series. This additional information has now been added to Figure 1’s caption.
- Figure 2. The interchangeable use of "LA4D" (Long‑Axis 4‑Dimensional) vs. "LA4C" (Long‑Axis 4‑Chamber) in the main text appears misleading. Thank you for pointing out this discrepancy. “LA4D” was a typographical error that has now been corrected, and LA4C is used throughout the revision.
- Figure 2. 2A, unlike 2C-2E, does not specify whether the data represents a single animal or an average. This specification has been added to the figure caption.
- Figure 3. 3C-E displays average signal intensity; hence, the sample size should be stated in the caption. This specification has been added to the figure caption.
- Figure 6. (B) description is stated twice. The second instance should be the description for (D). The figure caption has been corrected.
- Figure 7. 7B mentions capturing the image at 40x mag, but it is "shown at 0.8x magnification". This is ostensibly unclear and should be clarified. Is this display size reduction? Also, for "longitudinal strain (left), circumferential strain (middle plot), and radial strain (left plot)", should it be "radial strain (right plot)? Thank for your identifying this confusing wording and inconsistency in the figure caption. Clarity on magnification was been added and the strain-plot identification has been corrected.
Reviewer 3 Report
Comments and Suggestions for Authors
The authors should incorporate the following points into discussion:
- Although this protocol limits the total scan time to within 30 minutes, which is clearly shorter than traditional gated approaches, ECV is a biomarker of physiological status. Thus, anesthesia-induced changes in cardiovascular physiology remain a methodological limitation and may affect the accuracy of baseline ECV measurements.
- If the B1 field is inhomogeneous, systematic errors in T1 estimation may occur. Future work could incorporate B1 mapping and correction steps, or consider T1 mapping sequences that are less sensitive to B1 inhomogeneity (e.g., SASHA), although this would likely increase scan time and therefore requires careful trade-off considerations.
- The robustness of the SVD algorithm should be more comprehensively evaluated under different pathological conditions (e.g., advanced heart failure or pronounced motion), or more noise-tolerant phase-registration strategies should be developed.
Author Response
Reviewer 3 Comments & Responses
We thank the reviewer for the valuable comments and expert suggestions. We have made revisions according to the reviewer’s guidance, and believe this has greatly strengthened our manuscript. We answer the reviewer’s comments as below:
- Although this protocol limits the total scan time to within 30 minutes, which is clearly shorter than traditional gated approaches, ECV is a biomarker of physiological status. Thus, anesthesia-induced changes in cardiovascular physiology remain a methodological limitation and may affect the accuracy of baseline ECV measurements. Thank you, Reviewer, to point out this important point. Anesthesia is a limiting factor for all animal imaging. An additional paragraph at the end of the limitations section has been added to address this.
- If the B1 field is inhomogeneous, systematic errors in T1 estimation may occur. Future work could incorporate B1 mapping and correction steps, or consider T1 mapping sequences that are less sensitive to B1 inhomogeneity (e.g., SASHA), although this would likely increase scan time and therefore requires careful trade-off considerations. We appreciate the thoughtful consideration from the reviewer. You are absolutely right. B1 inhomogeneity can be a confounding factor for T1 measurement. Per your thoughtful guidance, we have added an additional section to the discussion “Future work and Additional Considerations” to address this point.
- The robustness of the SVD algorithm should be more comprehensively evaluated under different pathological conditions (e.g., advanced heart failure or pronounced motion), or more noise-tolerant phase-registration strategies should be developed. We agree with the reviewer. An additional section to the discussion has been added: “Future work and Additional Considerations” to address this point.
Round 2
Reviewer 3 Report
Comments and Suggestions for Authors
The revised version has addressed my concerns. I have no problem to endorse its publication. Thanks.